

# Electromagnetically induced transparency with Rydberg atoms across the Breit-Rabi regime

Julian B. Naber[1], Atreju Tauschinsky[2], H. Ben van Linden van den Heuvell[1] and Robert J. C. Spreeuw[1*]

**1** Van der Waals-Zeeman Institute, Institute of Physics, University of Amsterdam, Science Park 904, 1098XH Amsterdam, The Netherlands
**2** Department of Chemistry, University of Oxford, Chemistry Research Laboratory, 12 Mansfield Road, Oxford, OX1 3TA, UK

\* R.J.C.Spreeuw@uva.nl

## Abstract

We present experimental results on the influence of magnetic fields and laser polarization on electromagnetically induced transparency (EIT) using Rydberg levels of $^{87}$Rb atoms. The measurements are performed in a room temperature vapor cell with two counter-propagating laser beams at $480\,\mathrm{nm}$ and $780\,\mathrm{nm}$ in a ladder-type energy level scheme. We measure the EIT spectrum of a range of $ns_{1/2}$ Rydberg states for $n = 19-27$, where the hyperfine structure can still be resolved. Our measurements span the range of magnetic fields from the low field linear Zeeman regime to the high field Paschen-Back regimes. The observed spectra are very sensitive to small changes in magnetic fields and the polarization of the laser beams. We model our observations using optical Bloch equations that take into account the full multi-level structure of the atomic states involved and the decoupling of the electronic $J$ and nuclear $I$ angular momenta in the Breit-Rabi regime. The numerical model yields excellent agreement with the observations. In addition to EIT related experiments, our results are relevant for experiments involving coherent excitation to Rydberg levels in the presence of magnetic fields.



## 1 Introduction

Electromagnetically Induced Transparency (EIT), in essence a Fano-like interference between different excitation paths [1,2], opens up new possibilities for quantum information and non-linear optics, such as the creation of slowly propagating light [3] and photon storage and retrieval [4–6]. EIT in a three-level ladder (or "Ξ") scheme, involving an atomic Rydberg level, is also an attractive technique to gain spectroscopic information on Rydberg levels [7] or environmental influences [8–12], and can also be used for frequency stabilization of lasers [13]. Next to the multitude of spectroscopic applications, Rydberg EIT opens new paths for quantum information [14] and light-matter interaction, comprising single-photon sources [15], non-linear optics with single-photons [16], entanglement of light and atomic excitation [17,18], photon-photon interaction [19] and single-photon switches [20] and transistors [21]. Hot atomic vapor cells in conjunction with EIT [5, 22, 23] or Rydberg excitation are a well-established technique, where micrometer-sized vapor cells could provide low cost, scalable arrays of interacting qubits [24].

Here we describe EIT experiments in a room temperature vapor cell for the $^{87}$Rb $ns$-states with principal quantum number $n = 19 - 27$. We drive the transition from the $5s$ ground state level to Rydberg levels using a two-photon transition via the intermediate $5p$ level. The upper $5p - ns$ transition serves as the coupling transition, and we measure the effect on a weak, resonant probe laser tuned to the $5s - 5p$ transition. Despite the fact that our measurements are performed in a Doppler-broadened room-temperature vapor cell, we retrieve spectrally narrow EIT signals with a resolved Rydberg hyperfine splitting. Remarkably, the spectra change significantly already upon magnetic field variations of $\sim 0.1$ G.

It is known that the polarization of the light influences the spectrum [25, 26] through optical pumping effects [27]. A full description must consider the multi-level structure of the atom [28], typically the hyperfine and Zeeman substructure [28–30]. In order to explain our observations, we calculate the full density matrix for all 18 involved Zeeman levels by solving the optical Bloch equations (OBE). Fitting the solutions to our data involves averaging over the thermal velocity distribution, which is efficiently done on a supercomputer. We observe a strong influence on the spectra even when applying small magnetic fields ($\sim 0.1$ G), which we relate to the decoupling of the electronic $J$ and nuclear $I$ angular momenta. This finding is somewhat counter-intuitive, as one would expect that effect to be of major impact only at higher magnetic fields (Breit-Rabi regime). These results are important for all future applications using Rydberg excitation in the presence of magnetic fields. As an example, the so-called "magic field" of 3.23 G [31] is right in the Breit-Rabi regime for low-lying Rydberg states. At this field value the differential linear Zeeman shift between the ground state magnetic hyperfine sublevels $|F, m_F\rangle = |1, -1\rangle$ and $|2, 1\rangle$ vanishes. This makes this pair of levels a good candidate qubit with suppressed sensitivity to magnetic field noise. Hence, the findings in this paper are important in the context of magnetically trapped qubits.

## 2   Experimental setup

The heart of the experimental setup [see Fig. 1 (a)] consists of two laser beams at 480 nm (coupling beam) and 780 nm (probe beam), counter-propagating in a room temperature Rb vapor cell. The laser light is provided by two commercial diode lasers (TA-SHG Pro and DLpro, Toptica). Our experimental setup is similar to the one mentioned in Ref. [32], with the addition that we use a sideband-locking scheme to stabilize the lasers to a high-finesse Fabry-Pérot cavity. This procedure yields laser linewidths of less than 10 kHz and precise control over the absolute laser frequency [33]. Scanning of the laser frequencies is done by varying the corresponding sideband locking frequencies. The laser beams are spatially overlapped in the vapor cell, with a $1/e^2$ beam radius of 0.9 mm and 0.5 mm for the 480 nm and 780 nm light respectively. This configuration ensures that the probe light experiences a mostly uniform intensity distribution of the coupling light, and at the same time minimizes the effect of transit time broadening. Transit time broadening, due to the finite interaction time of Rb atoms at room temperature with the laser light, is estimated to be 400 kHz for the chosen value of probe beam radius. Typical laser powers are 10 $\mu$W for the probe and 150 mW for the coupling laser.

The vapor cell is 12 cm in length and is placed inside a 11 cm long coil consisting of 80 windings, introducing a near-homogeneous longitudinal magnetic field **B** along most of the vapor cell. Both vapor cell and coil are surrounded by a cylinder of mu-metal with a length of 175 mm and a diameter of 100 mm. We measure with a fluxgate magnetometer that the mu-metal reduces the parallel ambient magnetic field from 550 mG to 40 mG in the center, and 54 mG at the entrance plane of the cylinder. The magnetic field in the radial direction almost completely vanishes in the center.

Before taking EIT spectra, we fix the frequency of the probe laser at the $5s_{1/2}, F = 2 \rightarrow 5p_{3/2}, F' = 2$ transition of $^{87}$Rb by adjusting the sideband frequency of the locking. This frequency is referenced to Doppler-free absorption spectroscopy in an additional Rb vapor cell. We then scan the frequency of the coupling laser across the Rydberg states $ns_{1/2}, F'' = 1, 2$ for $n = 19 - 27$, where we can still distinguish the individual hyperfine levels [see Fig. 1(b)]. The frequency is scanned by stepping the locking sideband frequency, typically in equal steps of a few tens of kHz. After each step, we measure the transmission of the probe laser with a photo diode. An optical chopper in the coupling laser beam is used in combination with lock-in detection of the probe transmission to enhance the signal-to-noise ratio. We take one spectrum for each chosen magnetic field value inside the vapor cell.

## 3   Theoretical model

We investigate EIT in a configuration of four independent hyperfine levels as depicted in Fig. 1, consisting of the ground state $5s_{1/2}, F = 2$, the intermediate state $5p_{3/2}, F = 2$ and the Rydberg levels $ns_{1/2}, F = 1, 2$ for $n = 19 - 27$. As expected from earlier findings [25, 26], we observe that the EIT spectrum changes with different polarizations of probe and coupling laser. Therefore, we incorporate the substructure of magnetic Zeeman-levels for all the involved hyperfine states. Additionally, we measure a strong influence on the spectrum when applying a longitudinal magnetic field to the vapor cell. The changes are already noticeable for small magnetic fields of around 100 mG, and depend on the direction of the applied field. We therefore take into account the couplings and level shifts of magnetic sublevels leading to the Breit-Rabi diagram for the Rydberg manifold.

In Ref. [32] the spectrum of the two Rydberg hyperfine levels $ns_{1/2}, F = 1, 2$ for $n = 20 - 25$ is fitted by the sum of two individual solutions to the analytical model of a three level ladder system. In other references, including [27], the Zeeman substructure is accounted for by a sum

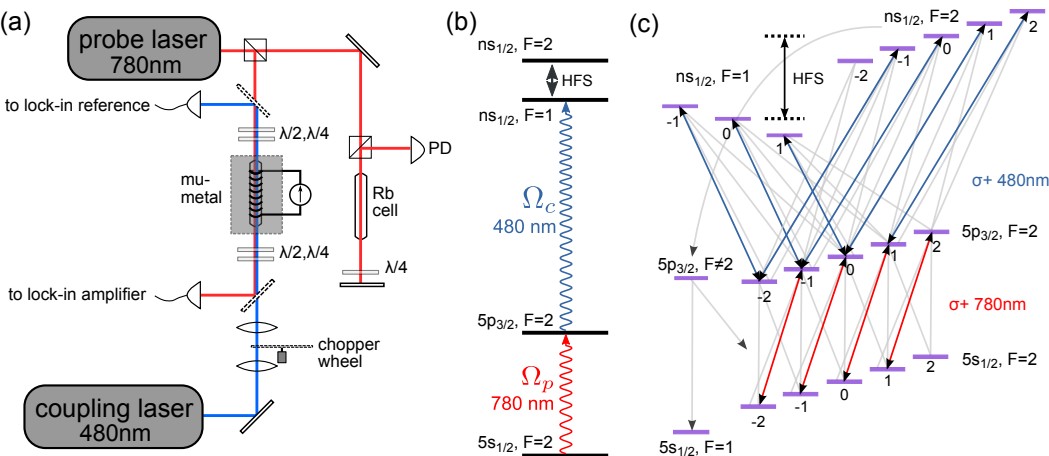

Figure 1: (a) Sketch of the experimental setup, showing the Rb vapor cell and the two diode lasers at 480 nm and 780 nm in a counterpropagating arrangement. The two laser beams are separated after traveling through the vapor cell by two dichroic mirrors, and detected by two photodiodes. The vapor cell is magnetically shielded by several layers of mu-metal, and surrounded by a coil with 80 windings and 11 cm in length. The 480 nm light is chopped by a wheel to generate a reference frequency for the lock-in detection amplifier. In addition, we perform Doppler-free saturation spectroscopy in another Rb vapor cell with the 780 nm laser. (b) EIT ladder scheme with the four involved hyperfine states of the ground, intermediate and Rydberg levels (HFS: Hyperfine-Splitting), and the probe and coupling laser. (c) Magnetic Zeeman levels $m_F$ of the ground state, the intermediate state and the Rydberg states involved in the ladder type EIT scheme (note: For the Rydberg level the $F, m_F$ states are only good quantum numbers in the limit of low magnetic fields). We show the excitation paths for the combination of $(\sigma^+, \sigma^+)$ polarization for the probe and coupling light respectively. The gray (light) lines show all considered decay paths for atomic populations in the excited states. The atomic levels $5s_{1/2}, F = 1$ and $5p_{3/2}, F \neq 2$ do not participate directly in the EIT ladder scheme, but they are populated by decay of atomic population.

over the involved levels for a given light polarization, weighted by the corresponding Clebsch-Gordan coefficients. Neither approach can explain the influence of the magnetic field that we see in our experiment. Therefore, we consider the full dynamics of the density matrix $\varrho$ of all the 18 Zeeman levels of the four hyperfine states depicted in Fig. 1(c). The atomic levels $5p_{3/2}, F = 0, 1, 3$ are only included indirectly as a decay channel for the atomic population in the Rydberg state, subsequently decaying to either $5s_{1/2}, F = 1$ or $5s_{1/2}, F = 2$. Atomic population decaying to $5s_{1/2}, F = 1$ is treated as loss, as these atoms no longer participate in the excitation dynamics.

Due to the geometry of our experiment (the laser beams propagate parallel to the magnetic field **B**), we can only achieve either $\sigma^+$ or $\sigma^-$ polarization in the quantization axis set by **B**. Hence, we limit our analysis to a combination of $(\sigma^+, \sigma^+)$ or $(\sigma^+, \sigma^-)$ polarization for probe and coupling laser [note: the cases $(\sigma^-, \sigma^-)$ and $(\sigma^-, \sigma^+)$ correspond to an inversion of the magnetic field].

We describe the dynamics of the system including the atom-light interaction, spontaneous decay and other decoherence effects by the master equation

$$\dot{\varrho} = \frac{i}{\hbar}[\varrho, H] + \mathscr{L}_{\text{decay}}(\varrho) + \mathscr{L}_{\text{deph}}(\varrho), \tag{1}$$

yielding a set of linear differential equations (Optical Bloch equations). Here, the Hamiltonian

$H$ describes the coherent part of the dynamics, whereas the Lindblad superoperators $\mathscr{L}_{\text{decay}}(\varrho)$ and $\mathscr{L}_{\text{deph}}(\varrho)$ describe effects causing decoherence.

## 3.1 The Hamiltonian

We decompose the Hamiltonian as $H = H_A + H_M + H_{AL}$, where the individual terms describe the field-free atomic energies, the magnetic energy and the atom-light interaction for all involved levels. As a basis set we choose the magnetic sublevels $F, m_F$ expressed in terms of the total angular momentum $F$ and the magnetic quantum number $m_F$. While $F, m_F$ are not good quantum numbers for the Rydberg levels, we find this basis nevertheless convenient.

The (magnetic-field free) atomic Hamiltonian is written using the dressed basis states and the rotating-wave approximation (RWA). It has a simple diagonal form (setting $\hbar = 1$),

$$H_A = \Delta_p P_{5s,F=2} - \Delta_c P_{ns,F=1} - (\Delta_c + A_{ns}) P_{ns,F=2}. \tag{2}$$

Here we defined the following symbols: $\Delta_p (\Delta_c)$ is the detuning of the probe (coupling) laser, the latter defined relative to the $F = 1, m_F = 0$ Rydberg state, $P_{5s,F=2}$ is a projection operator onto the $5s_{1/2}, F = 2$ subspace,

$$P_{5s,F=2} = \sum_{m_F=-2}^{2} |5s, F = 2, m_F\rangle\langle 5s, F = 2, m_F|, \tag{3}$$

and similar for the $P_{ns,F}$ projection operators. The $5p_{3/2}, F = 2$ intermediate level has been arbitrarily chosen as the zero of energy. Finally, $A_{ns}$ is the hyperfine splitting in the Rydberg level.

For the $5s$ and $5p$ subspaces the magnetic Hamiltonian $H_M$ is written as $H_M = g_F \mu_B F_z B$, with $F_z$ the $z$ component of the total angular momentum operator $\mathbf{F}$, and choosing the magnetic field as $\mathbf{B} = B\hat{z}$. In our basis set, this results in $H_M|F, m_F\rangle = g_F \mu_B m_F B|F, m_F\rangle$. An important aspect for the Rydberg states is that the atomic energies experience a transition from a linear energy dependency in $m_F$ at small magnetic fields to a decoupling of the magnetic quantum number $m_I$ and $m_J$ at high magnetic fields, called the Paschen-Back regime. The transition between these regimes, the Breit-Rabi regime, is shown for the example of $23s_{1/2}$ in Fig. 2(a). For the Rydberg levels, we write $H_M = g_S \mu_B S_z B + g_I \mu_B I_z B$ (as $J_z = S_z$ for the $ns_{1/2}$ states). Here $S_z$ and $I_z$ are the $z$ components of the electron spin $\mathbf{S}$ and nuclear spin $\mathbf{I}$. In the following we neglect the second, nuclear spin term. The first, electronic spin term has diagonal as well as off-diagonal matrix elements in the chosen $|F, m_F\rangle$ basis. The off-diagonal elements couple states of equal $m_F$ but unequal $F$,

$$\langle F, m_F | S_z | F', m_F\rangle = \sum_{m_s, m_I} m_s \langle F, m_F | s, m_s, I, m_I\rangle \langle s, m_s, I, m_I | F', m_F\rangle, \tag{4}$$

yielding $\langle 1, -1 | S_z | 2, -1\rangle = \langle 1, 1 | S_z | 2, 1\rangle = \sqrt{3}/4$ and $\langle 1, 0 | S_z | 2, 0\rangle = 1/2$. The diagonalization of $H_A + H_M$ in the Rydberg $ns$ subspace yields the Breit-Rabi diagram shown in Fig. 2(a). We find that these off-diagonal elements are crucial to accurately describe the measured EIT spectra. If we tentatively express the Rydberg Zeeman energy linear in $m_F$, we cannot reproduce our experimental observations. Remarkably, the off-diagonal elements contribute significantly already at small magnetic fields around $100\,\text{mG}$, which is much less than the hyperfine field ($\hbar A_{20s}/\mu_B \approx 5\,\text{G}$) and therefore far from the Paschen-Back regime.

The matrix elements of the atom-laser interaction Hamiltonian $H_{AL}$ are given by the usual products of a reduced dipole matrix element and a Clebsch-Gordan coefficient. For the $5s-5p$ transition, we can write the matrix elements of $H_{AL}$ as

$$\langle 5s(F = 2, m_F)| H_{AL} |5p(F = 2, m_F')\rangle = \frac{1}{2}\hbar\Omega_p \epsilon_q \langle 2, m_F, 1, q | 2, m_F'\rangle, \tag{5}$$

with $\epsilon_q$ the component of the laser amplitude with polarization $q = \pm 1$. Similar expressions apply for the $5p - ns$ transitions. In this case we write $\Omega_c$ for the Rabi frequency.

## 3.2 Dissipative terms

The second term in the sum of Eq. (1) accounts for the spontaneous decay and optical pumping. It can be written by means of the Lindblad superoperator $\mathscr{L}_{\mathrm{decay}}(\varrho)$ as

$$\mathscr{L}_{\mathrm{decay}}(\varrho) = \sum_{\{i,f\}} \left[ C_{fi}\, \varrho\, C_{fi}^{\dagger} - 1/2 \left( C_{fi}^{\dagger} C_{fi}\, \varrho + \varrho\, C_{fi}^{\dagger} C_{fi} \right) \right], \tag{6}$$

where $C_{fi} = \sqrt{\Gamma_{fi}}\,|f\rangle\langle i|$ is a quantum jump operator for the transition $i \to f$, with corresponding rate $\Gamma_{fi}$. The summation is performed over all allowed pairs $\{i,f\}$ of $F, m_F$ sublevels. The decay rate $\Gamma_{fi}$ is expressed as the product of the decay rate of the involved hyperfine-level ($\Gamma_{5p}$, $\Gamma_{ns,F=1}$ and $\Gamma_{ns,F=2}$) and the square of the corresponding Clebsch-Gordan coefficient.

For final states $f$ outside the considered subspace of 18 levels we omit the term $C_{fi}\varrho\, C_{fi}^{\dagger}$, which thus leads to loss of total atom population. For example, atomic population in the intermediate state $5p_{3/2}, F = 2$ can decay to either the $5s_{1/2}, F = 1$ or $F = 2$ ground state, where the former is treated as loss of atoms. As we treat atomic population decaying to $5s_{1/2}, F = 1$ as a loss mechanism, we omit the term $C_{fi}\varrho\, C_{fi}^{\dagger}$ in Eq. (6) for this level. For simplicity, we assume that atomic population in the Rydberg states predominantly decays to the $5p_{3/2}$ level. We further simplify the problem by assuming that the atomic population decaying to $5p, F \neq 2$ undergoes an immediate subsequent decay to either the $5s_{1/2}, F = 1$ or $F = 2$ ground state. This is justified by the fact that the $5p, F \neq 2$ levels are far off-resonant with respect to the probe laser, and that $\Gamma_{5p} \gg \Gamma_{ns,F=1}, \Gamma_{ns,F=2}$.

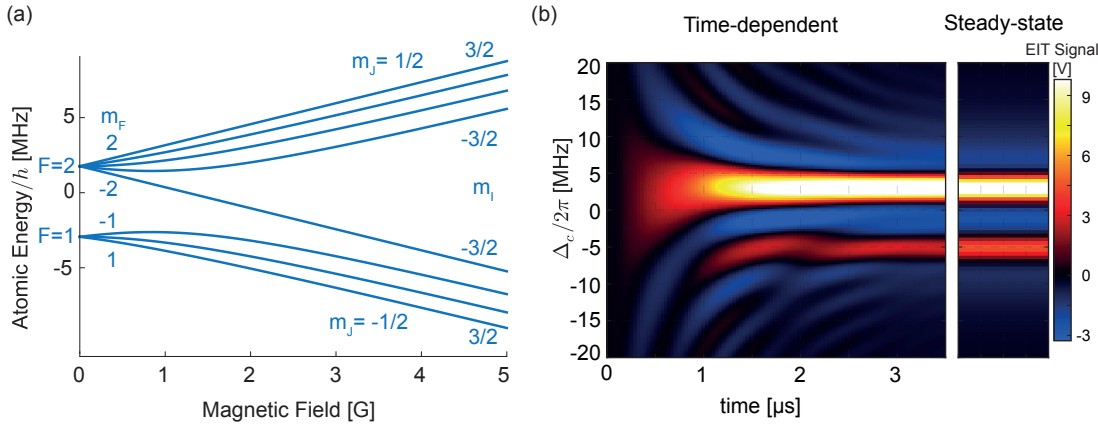

Figure 2: (a) Calculated atomic energies of the magnetic sublevels of $23s_{1/2}, F = 1$ and $F = 2$ in the Breit-Rabi regime. For small magnetic fields ($\ll 1\,\mathrm{G}$) the atomic levels of the hyperfine-states $F = 1, 2$ are labeled by the magnetic quantum numbers $m_F$ and shift linearly with $B$. For higher magnetic fields ($> 3\,\mathrm{G}$) the nuclear spin $I$ and the electron angular momentum $J$ decouple, and the magnetic levels group according to $m_J$. (b) Simulated EIT spectra to compare the time-dependent and the steady-state solutions of the Optical Bloch equations.

The third term $\mathscr{L}_{\mathrm{deph}}(\varrho)$ in Eq. (1) describes all dephasing effects, including the influence of the finite laser linewidth of the probe $\gamma_p$ and coupling $\gamma_c$ laser. For simplicity, we include additional broadening effects such as transit time broadening and collision-induced broadening

in $\gamma_c$. In this case, we express $\mathscr{L}_{\mathrm{deph}}$ as

$$\mathscr{L}_{\mathrm{deph}}(\varrho) = \sum_{k=p,c} \gamma_k \left[ C_k \varrho \, C_k^\dagger - 1/2 \left( C_k^\dagger C_k \varrho + \varrho \, C_k^\dagger C_k \right) \right], \tag{7}$$

where $C_p = -P_{5s} + P_{5p} + P_{ns}$ and $C_c = P_{5s} + P_{5p} - P_{ns}$ are expressed in terms of the projection operators as defined earlier.

## 3.3 Steady state solution and susceptibility

If we solve for the steady state ($\dot{\varrho} = 0$) of, for example, the system depicted in Fig. 1(c), we obtain the obvious result that the atomic population resides in the dark states $5s_{1/2}, F = 2, m_F = 2$ and $5s_{1/2}, F = 1$. Hence, this simple steady-state solution cannot explain our experimental data. In order to find an adequate description of the excitation dynamics, we follow two approaches: (1) Starting from an equal distribution among the ground state Zeeman levels, we calculate the time-dependent solution of the OBE, and evaluate it at the average time that an atom resides in the probe beam ($\tau \approx 3\,\mu s$ at room temperature), or (2), we assume constant fluxes of atoms leaving and entering the probe beam, the latter refilling the atomic population in the magnetic ground states. The flux into or out of the beam can in principle be estimated as $\Phi = (1/4)n\bar{v}A$, with $n$ the atom density, $\bar{v} = \sqrt{8k_B T/\pi m}$ the average thermal velocity and $A = \pi D L$ the surface area of a beam of diameter $D$ in a cell of length $L$. For the simulation we are merely interested in setting $\Phi \neq 0$, to ensure that the steady state is not a dark state. The precise value of $\Phi$ is then an overall multiplier to the amplitude of all simulated signals. Thus we describe the departure and arrival of atoms by adding $\partial \varrho / \partial t = (P_{5s,F=2} - \varrho)/\tau$ to the optical Bloch equations.

For the latter approach we obtain a steady-state solution with atomic population also being in non-dark states. Fig. 2(b) shows simulated spectra obtained for both approaches. It should be noted there that we subtract a background spectrum (with $\Delta_c$ being far off-resonant) from the time-dependent solution. We can conclude that both approaches yield similar results. As the second approach is closer to the experimental reality, we proceed with this one for the rest of this work.

The probe absorption is proportional to the imaginary part of the susceptibility $\chi$. We relate the susceptibility $\chi$ of the probe transition to the density matrix [1]. For the probe transition with polarization $q = \pm 1$, we look at the elements $\varrho_{ij}$ corresponding to a transition from a ground state $|g_i\rangle = |F = 2, m_F\rangle$ to an intermediate state $|e_j\rangle = |F' = 2, m_F + q\rangle$, with Clebsch-Gordan coefficient $c_{ij}$. We approximate the probe absorption as follows,

$$\mathrm{Im}(\chi) \propto \int_{-v_{\max}}^{v_{\max}} \sum_i c_{ij} \mathrm{Im}(\varrho_{i,j}) N(v) \, dv. \tag{8}$$

Here $N(v)$ is a one-dimensional Maxwell-Boltzmann velocity distribution for the atoms in the vapor cell at room temperature. The elements $\varrho_{ij}$ become velocity dependent through the Doppler shifts $\Delta_p \to \Delta_p^{v=0} - k_p v$ and $\Delta_c \to \Delta_c^{v=0} + k_c v$. We numerically evaluate the integral in Eq. (8) for a sufficiently large $v_{\max}$, effectively averaging our expression over the velocity distribution of the atoms.

## 3.4 Computational methods

We implement numerical solvers for both the time-dependent and the steady-state model using Fortran modules to solve the master equation [Eq. (1)], employing routines from the *odepack* library to solve the resulting system of complex differential equations. These Fortran modules are combined with a Python wrapper for the velocity-class integration of Eq. (8) as well as

for the loading of experimental data, fitting the model to experimental traces and storing the results. For a given experimental trace the measured data will be sampled for a fixed range of coupling frequencies using spline interpolation if necessary to gain control over the sampling density for numerical performance. We then call out to the Fortran solver to obtain solutions to the model on an appropriate grid of probe- and coupling frequencies. These are integrated in Python over a range of velocity classes, taking appropriate Doppler shifts into account. Finally the result is compared to the experimental trace. Fitting is performed using the *lmfit* routines in Python.

In fitting the experimental data we initially determine a magnetic field calibration based on the data for 20$s$ presented below in Fig. 3. This field calibration is used for all subsequent fits presented here. In fitting the data for a given principal quantum number $n$ and polarization, we always fit all traces (measured at different applied magnetic fields) with the same set of parameters and the field calibration obtained in the fit for 20$s$. We generally fit a linear combination of both the $(\sigma^+, \sigma^+)$ and the $(\sigma^-, \sigma^+)$ cases to account for imperfect polarization. The free parameters varied in the steady-state fits are the Rabi frequencies of the red and blue transitions, $\Omega_p$ and $\Omega_c$ respectively for both polarizations, the effective linewidths of these transitions $\gamma_p$ and $\gamma_c$, the hyperfine splitting of the state, the refilling rate for the ground-state as well as a global amplitude of the signal and an absolute frequency offset. This number of fitting parameters may seem rather large, however one set of parameters describes up to 41 individual traces (in Fig. 3). Furthermore, not all parameters are equally significant. Of primary interest are the hyperfine splittings, for which we find $A_{ns} \times n^{*3}/2\pi = 36.3(4)\,\text{GHz}$ [with $n^* = (n - \delta)$ the effective principal quantum number]. The fitted Rabi frequencies (given in the caption of Fig. 3) are consistent with the estimated intensities of the laser beams. The fitted effective linewidths $\gamma_c$, $\gamma_p$ were in the few 100 kHz range, which is plausible and difficult to check independently. The refilling rate and the global amplitude were essentially interchangeable.

The system of complex differential equations is large due to the 18 involved Zeeman levels. Calling the *odepack* library during the fitting procedure is therefore computationally intensive. When performing the velocity class integration necessary to obtain a single data point, we need to solve the system of equations for each velocity class separately. This further increases the computational complexity. In order to obtain results on acceptable timescales, we use the supercomputing capabilities of the Lisa Compute Cluster (as part of the SURFsara Research Capacity Computing Services). The fitting routine for a given Rydberg state and a given polarization is allocated to one node of the Lisa Cluster, which consists of 16 independent cores. Running the program for about 5 days on one node gives a sufficient amount of iterations to obtain acceptable fitting results. By employing different nodes for different states at the same time, we can evaluate the data in parallel.

## 4 Experimental results

Our measurements are based on acquiring the EIT signal at a specific detuning $\Delta_c$ of the coupling laser whilst keeping the probe laser at a constant frequency. Scanning the detuning $\Delta_c$ as described in Sec. 2 at a specific applied magnetic field value, we acquire a magnetic field dependent EIT spectrum. In order to verify that electric stray fields do not cause the observed changes to the spectrum, we temporarily introduced a vapor cell with electric field plates inside (not shown in Fig. 1) to measure the influence of electric fields. These plates allow for applying a near-homogeneous electric field (compare [32]) inside the cell. For small applied electric fields (a few V cm$^{-1}$) we do not observe a change in the spectral features besides an overall frequency shift due to the electric Stark effect.

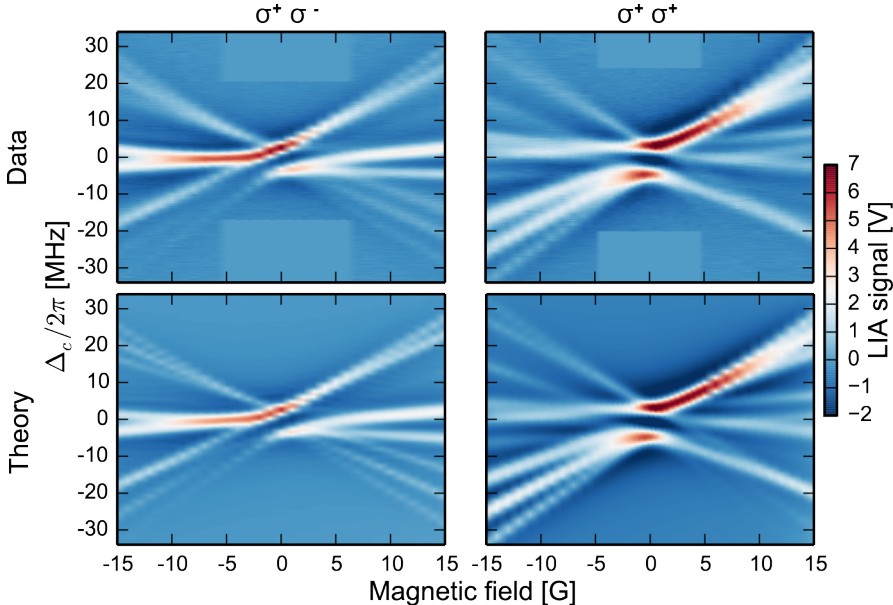

Figure 3: Measured (LIA: Lock-in amplifier signal) and simulated EIT spectra for the $20s_{1/2}$ Rydberg level for both the $(\sigma^+, \sigma^-)$ and $(\sigma^+, \sigma^+)$ combination of probe and coupling laser polarization. The density plot of the data shown consists of 31 and 41 individual EIT spectra, respectively, with a frequency resolution of 80 kHz in $\Delta_c$. As we have no absolute reference for the 0 MHz mark in the experiment, it is here chosen midway between the two two-photon resonances at zero field. Each spectrum is taken at a different magnetic field value ranging from $-15\ldots15$ G. For magnetic fields between $-5\ldots5$ G the detuning was scanned over a smaller range, $-20\ldots20$ MHz, because outside this range no spectroscopic features could be observed. The simulated data is based on the fitted theory parameters evaluated at the same magnetic fields and frequencies as the data. The fitted hyperfine splittings are 7.70 and 7.71 MHz, for $(\sigma^+, \sigma^-)$ and $(\sigma^+, \sigma^+)$, respectively. The fitted Rabi frequencies $(\Omega_p/2\pi, \Omega_c/2\pi)$ are $(14.7, 5.1)$ MHz and $(6.8, 5.1)$ MHz, respectively.

We aim at investigating the Breit-Rabi transition of the Rydberg states' magnetic sublevels, from a linear behavior in $m_F$ at low magnetic fields to a decoupling of $m_F$ into its components $m_I$ and $m_J$ at higher magnetic fields (the Paschen-Back regime). We probe this transition for the $20s_{1/2}$ Rydberg level by applying a range of magnetic fields from $-15$ G to 15 G and measuring EIT spectra. These spectra constitute the density plots shown in Fig. 3, which are based on measurements for either $(\sigma^+, \sigma^-)$ or $(\sigma^+, \sigma^+)$ probe and coupling laser polarization. The choice of either $\sigma^+$ or $\sigma^-$ polarized light leads to the simplest description of the system's dynamics, as the laser light polarization cannot have a component in the magnetic field direction (see Sec. 2). We verify for selected EIT spectra that the spectrum for the $(\sigma^-, \sigma^+)/(\sigma^-, \sigma^-)$ configuration closely resembles the one at $(\sigma^+, \sigma^-)/(\sigma^+, \sigma^+)$ after inverting the magnetic field. Hence, the resulting magnetic field dependence can be obtained by simply mirroring the data in Fig. 3 about the frequency axis. Furthermore, by creating an equal superposition of $\sigma^+$ and $\sigma^-$ polarization for both lasers, we obtain a spectrum which resembles a mixture of both data sets shown. Independent of these findings, we allow for a small admixture of the opposite polarization in the fitting procedure (see Sec. 3.4). This accounts for the fact that we always have imperfect polarizations in the actual experimental apparatus. For example, the change in polarization introduced by the waveplates in the optical setup before the vapor cell [compare Fig. 1(a)] is wavelength dependent (e.g. when changing between different $n$). Also, the glass cell itself might introduce further modifications of the laser polarization which

is difficult to predict.

Both data sets show a multitude of different lines, originating from the two hyperfine levels $F'' = 1$ and $F'' = 2$ of the Rydberg state, which are resolved at magnetic fields close to 0 G. In order to gain a qualitative understanding of the data, one can identify that two photons with $(\sigma^+, \sigma^-)$ and $(\sigma^+, \sigma^+)$ polarization lead to a change of $\Delta m_F = 0$ and $\Delta m_F = 2$, respectively. Thus, in the case of $(\sigma^+, \sigma^-)$ we expect the transition frequencies to stay roughly constant with increasing magnetic field, whereas for $(\sigma^+, \sigma^+)$ the transition frequencies are expected to increase with the applied magnetic field. Indeed, this expected behavior is visible in the data sets shown by the most pronounced lines in each plot. At higher magnetic fields ($> 5$ G) the frequencies of the observed experimental lines shift linearly with the applied magnetic field. This can be well understood in terms of the linear energy shift of the ground state $m_F$ levels, and the linear shift of the Rydberg state $m_J$ levels in the Paschen-Back regime [see Fig. 2(a)]. Hence, the transition frequency between these levels is also linear in the applied magnetic field. The multitude of different magnetic sublevels involved [compare to Fig. 1(c)] lead to a range of different transition frequencies, which show a different magnetic field dependence. This is reflected by the difference in slope of the experimental lines. It should be noted that the measured spectra are not a trivial reproduction of the simple Breit-Rabi diagram, as it also contains the magnetic field substructure of the ground and intermediate levels.

Besides the qualitative description, we also provide a theoretical account based on solving Eq. (1) for the system under investigation and using the fitting routine as described in Sec. 3.4. We show the theoretical result for both combinations of laser polarization in Fig. 3. Comparing the theoretical predictions and the actual data, we find that it matches very well for the full range of applied magnetic fields. All major experimental lines are reproduced, as are their relative strength and magnetic field dependence. Our model also describes the non-linear behavior in the Breit-Rabi regime at magnetic fields between 1...5 G equally well as the near linear behavior for magnetic fields in the Paschen-Back regime. Overall, the good agreement between measurement and theoretical simulation verifies our theoretical assumptions made in Sec. 3.

In order to examine the Breit-Rabi regime of the Rydberg magnetic sublevels in more detail, we investigate the response of the EIT spectrum to small changes in the applied magnetic field. Therefore, we acquire EIT spectra at nine equidistant magnetic field values in the range from $-0.8...0.8$ G. We present these spectra for the $19s_{1/2}$, the $21s_{1/2}$ and the $23s_{1/2}$ Rydberg level and different combination of probe and coupling laser polarization in Fig. 4. The $F'' = 1$ and $F'' = 2$ hyperfine levels are visible as two distinct peaks, separated by the hyperfine-splitting of the respective Rydberg state. The acquired spectrum for the $19s_{1/2}$ Rydberg state only shows a weak influence of the applied magnetic fields. The influence is much more pronounced for the $21s_{1/2}$ and $23s_{1/2}$ Rydberg levels. In the latter case we can observe an inversion of the relative peak height with changing the magnetic field polarization from negative to positive values.

Furthermore, we present the simulated EIT signal for the respective Rydberg states. Again, the simulation is based on fitting the result of Eq. (1) to the data set under investigation (see Sec. 3.4). As for the measurement in Fig. 4, the theoretical prediction closely reproduces the main features of the measured spectra as relative peak height and magnetic field dependence. The inversion of the relative peak height for the $23s_{1/2}$ state also appears in the simulated spectra. Given the excellent agreement with the simulation, this behavior can be well understood from the presence of off-diagonal terms as given by Eq. (4) in the magnetic Hamiltonian $H_M$. These terms result from the decoupling of the $J$ and $I$ quantum numbers of the Rydberg states in the Breit-Rabi regime, and introduce an effective mixing of the $F$ states. This effect increases with decreasing hyperfine-splitting, which explains the differences between the spectra of the $19s_{1/2}$ and $23s_{1/2}$ state. Hence, we can indirectly observe the Breit-Rabi transition in our spectrum, even at small magnetic field values.

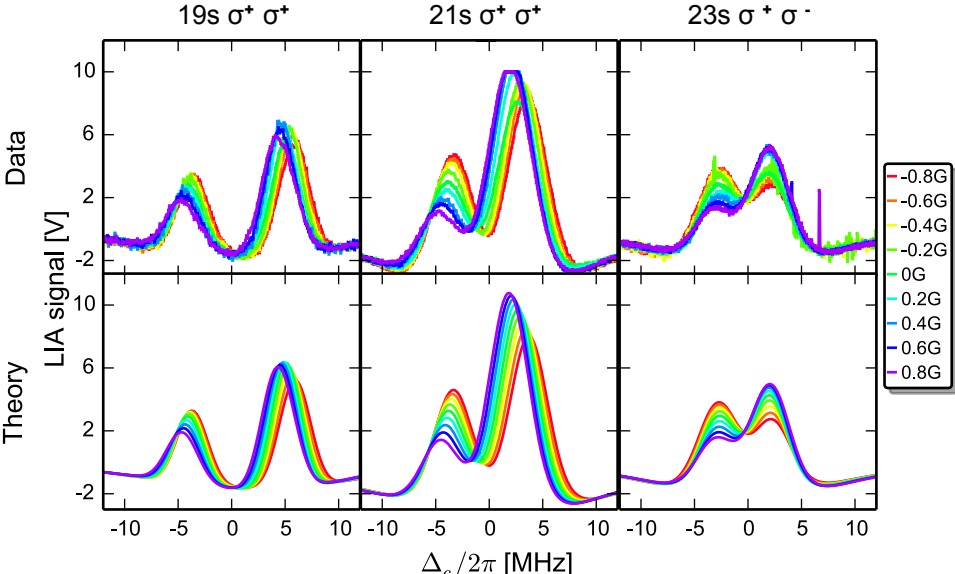

Figure 4: Measured (LIA: Lock-in amplifier signal) and simulated EIT spectra for the $19s_{1/2}$, the $21s_{1/2}$ and the $23s_{1/2}$ Rydberg level and different combination of probe and coupling laser polarization. The measured spectra are taken at nine equidistant magnetic field values in the range from $-0.8\dots0.8$ G. The data of one Rydberg state are fitted with a single set of parameters, resulting in the theoretical spectra shown beneath the respective Rydberg state. The fitted hyperfine splittings are 9.13, 6.34, and 4.68 MHz (left to right). The fitted Rabi frequencies $(\Omega_p/2\pi, \Omega_c/2\pi)$ are $(10.7, 3.8)$ MHz, $(11.0, 5.5)$ MHz, and $(7.3, 4.0)$ MHz. Note: the measured signal of the $21s_{1/2}$ state is truncated above 10 V by the data acquisition system. The sharp peaks in the $23s_{1/2}$ signals are spurious, due to electronic noise.

## 5 Discussion

Looking at the spectra in Fig. 3, it is obvious that the transition from low to high magnetic fields is not a simple reproduction of the Breit-Rabi diagram of the Rydberg levels as shown in Fig. 2(a). The reason is that the spectrum is also influenced by the level shifts of ground and intermediate states' Zeeman substructure, optical pumping effects and the residual Doppler-broadening. However, the spectrum clearly reproduces the selection rules introduced by the laser light polarization, and shows that the high field behavior is a linear function of the applied magnetic field. This is a direct result of the Paschen-Back regime for the Rydberg levels (linear in $m_J$) and the linear energy shift of the ground state levels (in $m_F$). A remarkable observation is that magnetic fields, small compared to the hyperfine field $A_{ns}/\mu_B$, strongly influence the spectra. This influence increases with decreasing hyperfine splitting $A_{ns}$ of the Rydberg levels, as can be seen by comparing the $n=19$ and $n=23$ Rydberg level in Fig. 4. For $n=23$ (and also for $n=24-27$) the change in magnetic field ($-0.8\dots0.8$ G) leads to a complete inversion of the relative height between the peaks attributed to $F''=1$ and $F''=2$. As discussed earlier we can attribute this to the influence of the off-diagonal elements in Eq. (4), which are a direct consequence of the decoupling of the total angular momentum $F$ into the components $J$ and $I$ in the Breit-Rabi regime. We verified this by calculating the corresponding spectra based on a model where the Rydberg states shift linearly in energy with $m_F$. The result did not reproduce the observed change in peak height, but solely predicts a frequency shift of the total spectrum. This shift is observed for the spectrum at $n=19$, where the hyperfine splitting is relatively large ($A_{19s}/2\pi\approx9$ MHz) so that the influence of the off-diagonal elements is less pronounced.

Despite the spectrum's complexity [compared to Fig. 2(a)], it is nevertheless possible to understand our results quantitatively. While we cannot simply extract the hyperfine splitting $A_{ns}$ of the Rydberg levels at $B = 0$, in our fitting routine, we use $A_{ns}$ as a fitting parameter for the complete data set at a given $n$. For the rescaled hyperfine splittings we find $A_{ns} \times n^{*3}/2\pi = 36.3(4)\,\text{GHz}$, similar to [32], but with slightly less scatter. In [32] EIT signals were fitted by the sum of two individual solutions to the analytical model of a three-level ladder system. The resulting (scaled) hyperfine splittings varied by about 3 percent.

Our measurements also show that precise values for the Rydberg hyperfine splittings can be obtained in room-temperature vapor cells. There are several options to further improve our measurements in future experiments. The magnetic shielding can be improved by embedding the vapor cell in a longer and narrower, mu-metal cylinder. Better magnetic field control is possible using a longer solenoid producing more homogeneous magnetic fields. A reduction of the number of fitting parameters appears feasible, as we found that the overall amplitude and refilling rate are interchangeable, and the red laser linewidth could essentially be fixed. The use of wider laser beams would reduce the influence of transit time broadening. Better control of the laser light polarization is also still possible, for example using in-situ measurement with a polarimeter.

# 6 Conclusion

Our measurements show that the EIT spectrum for the $ns_{1/2}$ Rydberg states with $n = 19 - 27$ is strongly influenced by the presence of small magnetic fields ($< 1\,\text{G}$) (see Fig. 4). Furthermore, the polarization of the involved laser beams changes the measured spectrum strongly (see Fig. 3). We investigate the EIT spectrum of the $20s_{1/2}$ Rydberg state for a wide range of magnetic field values (Fig. 3), showing a transition from two resolvable hyperfine levels to a multitude of lines with a linear frequency scaling. The experimental observations are well reproduced by the theoretical approach provided in Sec. 3. Our theoretical model accounts for the multi-level structure of the $5s_{1/2}, F=2$ ground state, the $5p_{3/2}, F'=2$ intermediate state and the two Rydberg states $ns_{1/2}, F''=1, 2$. An essential part of the modeling is also the averaging over the thermal velocity distribution in the vapor cell. A crucial aspect for the Rydberg states is the decoupling of the $F$ angular momentum into its components $I$ and $J$ in the Breit-Rabi regime. From the measurements in Fig. 3 we can retrieve the Rydberg states' behavior, both at small magnetic fields and in the Paschen-Back regime, where the magnetic sublevels group according to their $m_J$ quantum number (see Fig. 2). The behavior of the magnetic sublevels in the Breit-Rabi regime also accounts for the strong changes observed in the spectrum for the magnetic fields below $1\,\text{G}$ presented in Fig. 4. While we cannot resolve individual magnetic sublevels in the measurements at low magnetic fields, we can still clearly identify their influence on the spectrum, based on the excellent agreement with our theoretical model.

This sensitivity for weak magnetic fields makes it important to have a detailed understanding in a variety of applications of EIT in thermal vapors. Examples of such applications include photon storage and retrieval, nonlinear optics, the generation and manipulation of single-photons, quantum information science, Rydberg polaritons, etc. [3–5, 5–24]

# Acknowledgments

We would like to thank Bob Rengelink, Jannie Vos and Jana Pijnenburg for their contribution to the experimental apparatus. The numerical simulations were carried out on the Dutch national e-infrastructure with the support of SURF Cooperative. We thank SURFsara (www.surfsara.nl)

for the support in using the Lisa Compute Cluster. Our work is financially supported by the Foundation for Fundamental Research on Matter (FOM), which is part of the Netherlands Organisation for Scientific Research (NWO). We also acknowledge financial support by the EU H2020 FET Proactive project RySQ (640378). JN acknowledges financial support by the Marie Curie program ITN-Coherence (265031).

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
