# Peer review of "Electromagnetically induced transparency with Rydberg atoms across the Breit-Rabi regime"

_SciPost Physics, doi:SciPost Phys. 2, 015 (2017)_

## Round 2 · Referee Report · Anonymous (Referee 1) · 2017-2-19

Strengths

1 - Motivation and description of science.
2 - Sophisticated theory experiment comparison.

Weaknesses

1 - Discussion of results for fitting parameters.

Report

The manuscript describes an experimental and theoretical study of Rubidium Rydberg-EIT spectra in a room temperature vapor cell.
EIT is realized in a ladder configuration, with the stable ground state energetically at the bottom, a fast decaying excited state in the middle and a slowly decaying (metastable) Rydberg state at the top. The frequency of the upper transition laser is scanned across two-photon resonance and the light absorption/transmission features are recorded. Of central interest is the magnetic field dependence due to Zeeman shifts of the atomic levels involved.

The experimental results are compared with theoretical calculations involving the density matrix for 18 involved atomic states, including
decoherence and dephasing effects. Good agreement between experiment and theory is found when a large number of parameters in the theoretical model are fit to match the experimental data. The central conclusion of the present work, is that deviations from the weak field Zeeman effect towards decoupled nuclear and electron spins are essential to understand the experimental results and that such effects occur already at somewhat weaker magnetic fields than one would intuitively expect.

The results appear relevant to a large body of research on Rydberg EIT, the article is well written and the extent to which theory seems to have addressed all relevant experimental features impressive, despite the large number of fitting parameters in the model (see below). I would thus recommend publication of the article, after the following questions are addressed:

Requested changes

(1) In places the authors could make their manuscript more self contained with minimal effort:
..."the famous “magic-field” of 3.23 G [31]" ... Please expand this in 1-2 sentences explaining the relevance.
..."we find a spread of values similar to those in [32]"... Please describe in 1-2 sentences those results of [32] relevant here.

(2) My primary concern is the large number of fitting parameters in the model [see page 10, second last paragraph],
and the limited discussion of these afterwards. Of primary interest are the hyperfine splittings. There is a discussion of these at the top of page 16, which should be clarified somewhat.
Are the results of the fit as expected? The paragraph reads like this is rather not the case. Would it then not be better to take these values from theory
to reduce the number of free fitting parameters? I reckon the text should read "rescaling the results as $A_{ns} \times {n^*}^3$ (see [32] and scaling of Hyperfine-splitting with n^{-3}.
Also for the remaining fitting parameters I think at least a rough consistency check with theoretical expectations or estimates should be given.

(3) Caption Fig. 3: "A part of the data between −5...5G is omitted as it does not contain relevant spectroscopic features. ..."
I would welcome at least a footnote explaining what features were omitted and why it is certain that they are not relevant (to maybe some other question...)

(4) Finally, I am wondering if the authors have considered replacing the Master equation propagation with a stochastic wavefunction
in the quantum-jump monte-carlo formalism [see e.g. Molmer et al. JOSA B Vol. 10, pp. 524 (1993)]. This could possibly be much more efficient computationally. [This is not a requested change, just a suggestion and question out of interest]

  • validity: high
  • significance: good
  • originality: good
  • clarity: high
  • formatting: good
  • grammar: excellent

Author:  Robert Spreeuw  on 2017-03-09  [id 108]

(in reply to Report 1 on 2017-02-19)
Category:
answer to question

We thank the referee for her/his knowledgeable comments. We believe that the changes as detailed below would clarify the issues raised.

(1)
We follow this suggestion, and propose to expand
”the famous “magic-field” of 3.23 G [31]" to:
“...the famous “magic field” of 3.23 G [31] is right in the Breit-Rabi regime for low-lying Rydberg states. At this field value the differential linear Zeeman shift between the ground state magnetic hyperfine sublevels $\vert F,m_F\rangle=\vert 1,-1\rangle$ and $\vert 2,1\rangle$ vanishes. This makes this pair of levels a good candidate qubit with suppressed sensitivity to magnetic field noise. Hence, the findings in this paper are important in the context of magnetically trapped qubits.”

..."we find a spread of values similar to those in [32]"...
We propose to replace the entire two sentences, starting with “Rescaling…” plus the next sentence by:

“For the rescaled hyperfine splittings we find $A_{ns}\times n^{*3}/2\pi=36.3(4)\,\text{GHz}$, similar to [32], but with slightly less scatter.
In [32] EIT signals were fitted by the sum of two individual solutions to the analytical model of a three-level ladder system. The resulting (scaled) hyperfine splittings varied by about 3 percent. ”

(2) The hyperfine splittings are indeed of primary interest. We do not know of any accurate theoretical value to fix it to. Extrapolation from other experimental results is possible, see also [32] (insert to Fig. 3). See also our reply to comment (1)
The remark about rescaling the $A_{ns}$ values on page 16 was indeed incorrect in the manuscript. We thank the referee for finding this, will correct it, and add the value for the scaled hyperfine splitting.

Concerning the number of fitting parameters, we propose to add the following text to the last full paragraph on page 10:

“This number of fitting parameters may seem rather large, however one set of parameters describes up to 41 individual traces (in Fig. 3). Furthermore, not all parameters are equally significant. Of primary interest are the hyperfine splittings, for which we find $A_{ns}\times n^{*3}/2\pi=36.3(4)\,\text{GHz}$ [with $n^* = (n − \delta)$ the effective principal quantum number].
The fitted Rabi frequencies (given in the caption of Fig. 3) are consistent with the estimated intensities of the laser beams. The fitted effective linewidths $\gamma_c$, $\gamma_p$ were in the few $100\,\text{kHz}$ range, which is plausible and difficult to check independently. The refilling rate and the global amplitude were essentially interchangeable.”

(3) This formulation was confusing. We would like to reformulate it as follows:

“For magnetic fields between -5…5 G the detuning was scanned over a smaller range, -20 … 20 MHz, because outside this range no spectroscopic features could be observed. “

(4) This is indeed a very interesting suggestion. We did not use this method so far, because we were better set up to solve the master equation.

---

## Round 2 · Referee Report · Anonymous (Referee 2) · 2017-3-14

Strengths

1- quantitative theoretical understanding of Rydberg-EIT spectra 2- very clearly written

Weaknesses

1- lacking discussion of the implications for future experiments

Report

J. B. Naber et al present a detailed experimental and theoretical study of electromagnetically induced transparency resonances involving Rydberg states in a thermal vapour cell, including the effects of magnetic fields. This has relevance for potential applications of Rydberg states as precision sensors and future applications for quantum optics and quantum information science.

Overall I find the paper very clear on its exposition of the results. They obtain very good agreement with experimental spectra by fitting a model based on the solution of the optical Bloch equations including all relevant atomic sublevels. In my view they find nothing particularly unexpected, but provide a good demonstration that EIT spectra in thermal vapour cells can be quantitatively understood. As a general recommendation I would like to see the authors make a better effort to emphasise the implications of their findings for future experiments. For example, it is pointed out that the fitted value of the Rydberg-state hyperfine splitting takes a spread of values due to the difficulty in fitting such unconstrained (multidimensional) problems. I would like to know how this can be improved in order to obtain more accurate spectroscopic information which might be relevant for future experiments (e.g. with less common atomic species). The authors should also conclude on how their findings might impact future applications for Rydberg based sensors or single photon sources. Assuming this can be done, I have no hesitation on recommending publication in SciPost.

Requested changes

I. Introduction 1- At first sight, solving the 18 level OBEs would not require the resources of a supercomputer. Perhaps it would be fairer to mention here that you fit the results of an 18-level OBE simulation to the data, including averaging over the thermal velocity distribution, which is efficiently done on a supercomputer. 2- I found the sentence about the "decoupling influencing the spectra ... (Breit-Rabi regime), even at small magnetic fields ... (Zeeman regime)" confusing. Perhaps remove the reference to the Breit-Rabi regime. 3- I would remove the adjective "famous" when referring to the magic field of 3.23 G. For non experts it would be best to define what is specifically magic about 3.23 G

II. Experimental setup 4- Fig 1. typo "mu-metall" -> "mu-metal" 5- Fig 1 caption. The two laser beams are detected by two photodiodes? 6- Fig 1 caption. The gray (light) lines show all considered decay paths rather than "all possible decay paths". The 5p_{1/2} state, and other Rydberg states populated by black-body decay are not included for example.

III. A. The Hamiltonian 7- paragraph 1. Fix two typos when referring to $m_F$ 8- paragraph 3. It would be helpful to define $F_z$, $s_z$ and $I_z$ (as spin operators) 9- paragraph 3. $H_M|F,m_F\rangle$ (missing comma)

III. B. Dissipative terms 10- Fig 2. caption. The states referred to in the first sentence should be $23s_{1/2},F=1$ and $23s_{1/2},F=2$

IV. Experimental results 11- paragraph 1. What is the additional vapor cell with incorporated electric field plates? Is this a third cell not depicted in Fig 1? It wasn't clear to me how this is used to exclude the influence of stray electric fields inside the mu-metal shielded cell. 12- fig 3 caption. "The choice of 0 MHz is arbitrary". Surely it is not arbitrary in theory. Should this be "0 MHz corresponds to the zero field two-photon resonance position"? 13- I would like to see an additional paragraph detailing how the best fit parameters compare with independent estimates, e.g. for the Rabi frequencies, the polarization extinction ratio, the re-population rate, etc.

V. Discussion 14- I recommend the discussion be more oriented to the implications of this study for future experiments. This would change the emphasis from simply reporting an experiment to setting a path forward, e.g. for technological applications of Rydberg-EIT (see general comments above).

  • validity: high
  • significance: good
  • originality: good
  • clarity: high
  • formatting: excellent
  • grammar: excellent

Author:  Robert Spreeuw  on 2017-03-27  [id 111]

(in reply to Report 2 on 2017-03-14)
Category:
answer to question

We thank the referee for a careful and detailed account. We implemented the requested changes as detailed below:

I. Introduction 
1- We agree with this remark. We rephrase “… on a supercomputer” as follows: “. Fitting the solutions to our data involves averaging over the thermal velocity distribution, which is efficiently done on a supercomputer.”

2- We replaced the sentence “From the analysis … “ by “We observe a strong influence on the spectra even when applying small magnetic fields (~ 0.1 G), which we relate to the decoupling of the electronic J and nuclear I angular momenta. This finding is somewhat counter-intuitive, as one would expect that effect to be of major impact only at higher magnetic fields (Breit-Rabi regime).”

3- we replace the adjective “famous” by “so-called”. The explanation why this field value is special has been dealt with in our reply to the earlier Report 85.

II. Experimental setup 
4- Corrected this typo

5- Fig 1 caption: “… a photo diode” replaced by “… two photodiodes”
 6- Fig 1 caption: “… possible decay paths… ” replaced by “… considered decay paths… ”

III. A. The Hamiltonian 
7- paragraph 1. Replaced $mF$ by $m_F$, twice

8- paragraph 3. Added definitions for the spin operators $F_z$, $S_z$ and $I_z$

9- paragraph 3. $H_M|F,m_F\rangle$ (comma added)

III. B. Dissipative terms 
10- Fig 2. caption. Corrected this typo, now referring to states $23s_{1/2},F=1$ and $F=2$

IV. Experimental results 
11- paragraph 1. Replaced the sentence “To exclude a possible … plates” by: “In order to verify that electric stray fields do not cause the observed changes to the spectrum, we temporarily introduced a vapor cell with electric field plates inside (not shown in Fig. 1) to measure the influence of electric fields.”

12- fig 3 caption. Replaced “The choice of 0 MHz is arbitrary" by: “As we have no absolute reference for the 0 MHz mark in the experiment, it is here chosen midway between the two two-photon resonances at zero field.”

13- See very similar comments in the earlier “Report 84” and our response there.

V. Discussion
 14- We added text in the discussion and the conclusion.

Discussion: - sentence at the beginning of 2nd paragraph: “Despite the spectrum's complexity [compared to Fig. 2(a)], it is nevertheless possible to understand our results quantitatively.” - added a 3rd paragraph about possible improvements

Conclusion: - halfway in 1st paragraph, added a sentence: “An essential part of the modeling is also the averaging over the thermal velocity distribution in the vapor cell.” - at the end, added a few lines about the implications of this work for other experiments.

---

## Editorial Decision

published